# Evaluation of Surface Structure and Morphological Phenomena of Caucasian Virgin Hair with Atomic Force Microscopy

**DOI:** 10.3390/medicina60020297

**Published:** 2024-02-09

**Authors:** Karolina Krawczyk-Wołoszyn, Damian Roczkowski, Adam Reich

**Affiliations:** 1Doctoral School, University of Rzeszow, 35-959 Rzeszów, Poland; karolinakrawczyk10@wp.pl; 2Department of Dermatology, Institute of Medical Sciences, Medical College of the Rzeszow University, 35-959 Rzeszów, Poland; droczkowski99@gmail.com

**Keywords:** atomic force microscopy, atomic force microscope, AFM, dermatology, hair, human hair, diseases of the hair

## Abstract

*Background and Objectives:* Atomic force microscopy (AFM) as a type of scanning microscopy (SPM), which has a resolution of fractions of a nanometer on the atomic scale, is widely used in materials science. To date, research using AFM in medicine has focused on neurodegenerative diseases, osteoporosis, cancer tumors, cell receptors, proteins and the DNA mismatch repair (MMR) system. Only a few small studies of hair imaging have been conducted, mostly in biotechnology or cosmetology. Thanks to the possibilities offered by AFM imaging, dermatologists can non-invasively assess the condition of hair and its possible disorders. Our goal was to capture images and microscopically analyze morphological changes in the surface of healthy hair. *Materials and Methods:* In this study, three to five hairs were collected from each person. Each hair was examined at nine locations (0.5; 1.0; 1.5; 2.0; 3.5; 4.5; 5.5; 6.5 and 7.0 cm from the root). At least 4 images (4–10 images) were taken at each of the 9 locations. A total of 496 photos were taken and analyzed. Metric measurements of hair scales, such as apparent length, width and scale step height, were taken. *Results:* This publication presents the changes occurring in hair during the natural delamination process. In addition, morphoological changes visualized on the surface of healthy hair (pitting, oval indentations, rod-shaped macro-fibrillar elements, globules, scratches, wavy edge) are presented. A quantitative analysis of the structures found was carried out. *Conclusions:* The findings of this study can be used in further research and work related to the subject of human hair. They can serve as a reference for research on scalp and hair diseases, as well as hair care.

## 1. Introduction

Based on previous studies of hair carried out using scanning electron microscopy (SEM) and transmission electron microscopy (TEM), we have knowledge of hair fiber structure at the microscale, although there are still many unanswered questions in this area. The introduction of atomic force microscopy (AFM) as an alternative method in dermatology research can have great potential in the diagnosis and treatment of various skin and hair disorders [1].

Human scalp hair consists of a central medulla, then cortex and thin layers of cells (cuticle) surrounding it from the outside. Cuticle cells (scales) overlap each other (like tiles on a roof). The casing consists of an average of 5–10 cuticle cells. Each scale is typically 0.3–0.5 μm thick, has an apparent length of 5–10 μm and is separated from the underlying scale by a cell membrane complex [2]. To date, the internal structure and chemical composition of cuticles has been investigated and described using transmission electron microscopy (TEM) of transverse sections of hairs. As a result, we know that each cuticle cell consists of seven layers that differ in composition and tribological properties. These layers are as follows (from the most superficial to the deepest): the upper β-layer, the A-layer, the exocuticle, the endocuticle, the inner layer, the lower β-layer and the δ-layer. The lower β-layer and δ-layer of one cuticle, together with the upper β-layer of the scale below it, form the cell membrane complex (CMC). CMC is most likely one of the sites of delamination of the cuticle during mechanical wear and tear [3].

Despite some advantages, SEM and TEM also have limitations in terms of accurately visualizing spatial architecture of the hair surface at the nanoscale and the tribological properties of hair, which AFM makes possible. Hair fibers and cells have already been preliminarily examined by AFM in several studies. The researchers studied surface morphology, hair’s step heights and scale dimensions in healthy, virgin hair. In addition, they tested the tribological properties of hair [4,5] and mainly focused on the effects of external factors on the above-mentioned hair characteristics. They tested cosmetic preparations for hair care such as conditioners and shampoos, as well as the effects of cosmetic procedures such as coloring and perming [2,6,7,8,9].

The invention of AFM by Binnig et al. in 1986 laid the groundwork for imaging atomic structure. AFM is increasingly being used in scientific research, as evidenced by the fact that the number of citations for the inaugural article is now at approximately 23,000 [10]. The AFM technique is constantly improving and is a forward-looking scientific tool. Extending the functions of AFM with other techniques (such as spectroscopy) extended its utility. Various AFM modifications already exist, for example, single-cell force spectroscopy (SCFS), single-molecule force spectroscopy (SMFS), chemical force microscopy (CFM) and AFM-infrared spectroscopy (AFM-IR) [11,12,13].

In this study, an attempt was made to find such parameters of healthy hair, which can be helpful in a non-invasive, molecular, rapid assessment of the condition of hair and its possible abnormalities. It was proposed to create a characterization of healthy hair based on AFM images, which can be used by dermatologists to examine the hair of patients with various hair diseases. There are few studies focusing on this topic, and the AFM imaging results obtained in them are of inferior quality. The following publication presents the data on this topic more practically so that an attempt can be made to standardize the procedure for assessing the condition of hair for medical purposes. This study is based on a carefully selected control group of dermatologically examined healthy subjects by a dermatologist. Results from hair sections in 0.5–1 cm increments are presented, which is quite accurate for the time-consuming nature of this technique. This work was extended to present a hair diagram that dermatologists could use in their daily work as a reference for future studies on their patients’ diseased hair.

## 2. Material and Methods

### 2.1. The Objective of the Study

To date, few medical studies have been conducted on AFM imaging of hair diseases. This study demonstrates the appearance of healthy hair under AFM, which can be used for further research and work related to the subject of human hair. The goal of this study was to find the parameters of human healthy hair and map them to quickly assess its condition.

### 2.2. Subjects

The study group consisted of 10 healthy volunteers, 9 women and 1 man, with a mean age of 43.8 years with SD (Standard Deviation) = 16.19. Three to five virgin hairs, which had not been treated with any dye or other harmful treatments for at least 3 months, were taken from each subject. None of the included subjects suffered from any skin or hair disease.

### 2.3. Hair Preparation

The clean hairs were pulled out along with their roots and subsequently cut vertically with a scalpel into 3 cm long sections. They were then attached to a microscope slide using a translucent tape.

### 2.4. Atomic Force Microscopy (AFM)

AFM was performed using a DriveAFM microscope with CX controller (Nanosurf, Liestal, Switzerland) under atmospheric conditions (in air) at room temperature. Imaging was performed in contact mode using PPP-FMAuD-10 tips (NANOSENSORS™, Neuchatel, Switzerland) mounted on cantilevers with a constant force of 1.9 N/m and frequency of 75 kHz. Each hair was imaged at a distance of 0.5, 1.0, 1.5, 2.0, 3.5, 4.5, 5.5, 6.5 and 7.0 cm from the root. The scanning range covered an area from 40 × 40 µm to 3 × 3 µm. The scanning velocity was 0.5 s/line. The resolution of the images was 512–1024 points/line.

At least 4 images were taken at each location at different dimensions of the field. The images were computer-processed using Nanosurf CX ver 3.10.3.7. The AFM tip was centered to be on top of the hair fiber upon contact with the cuticle surface. The AFM tip scanned the hair perpendicular to the longitudinal axis of the fiber. This reduced errors due to the AFM tip hitting the sides of the hair.

### 2.5. Method of Taking Measurements and Analyzing Images Obtained with Atomic Force Microscopy (AFM)

At each of the 9 locations (0.5, 1.0, 1.5, 2.0, 3.5, 4.5, 5.5, 6.5 and 7.0 cm from the root), at least 4 images (4–10 images) were taken. Finally, a total of 496 photos were collected and analyzed.

Line profiles were created by averaging data from 10 adjacent scan lines. Line profile creation, 3D visualizations, image processing and all metric measurements (length, width, and scale step height—Figure 1) were performed using Gwyddion (64 bit) software ver 2.63.

Metric measurements were taken on 30 × 30 µm images from each location and from each subject. A total of 729 scales were evaluated. *Scale length* refers to the dimension parallel to the longitudinal axis of the hair. Due to the tile-like arrangement of the scales, only the portion of the scales available for examination was measured. *Scale width* refers to the transverse dimension to the longitudinal axis of the hair. To measure *scale deviation*, a line profile of adjacent scales was made. Lines were run mostly perpendicular to the free end of each scale. From the resulting profile, the height difference of each step was measured.

**Figure 1 medicina-60-00297-f001:**
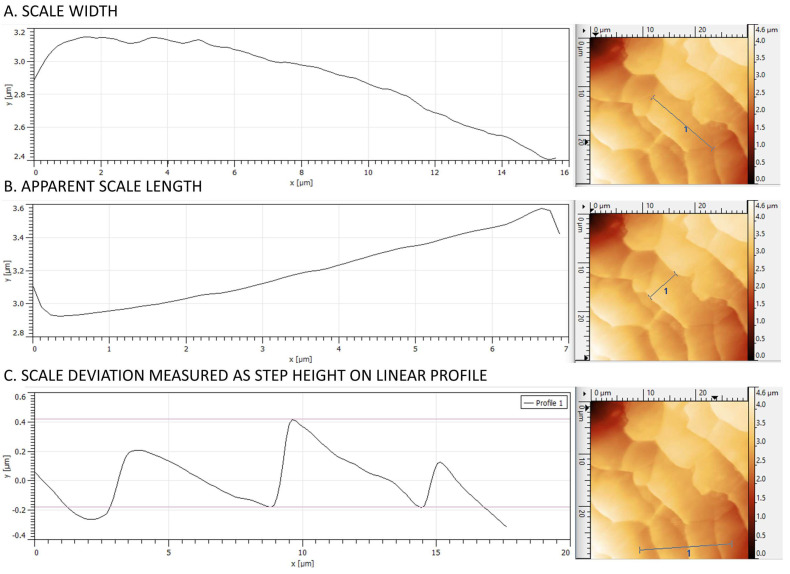
Parameters analyzed with AFM in terms of hair scale (*Z*-axis images).

The evaluation of features indicative of the natural process of hair delamination (striated surface, endocuticle, smooth surface, cortex, ghost signs, broken edges of scales, shape of edges) was performed in a quantitative and semi-quantitative manner. From the photos, (1) the number of scales with striated surface, (2) the number of scales with smooth surface and (3) the number of cavities corresponding to ghost signs were counted. Then, the above parameters were grouped on a scale from 0 to 5 as in Table 1.

Parameters such as (1) broken edges of scales, (2) shape of edges and (3) endocuticle were evaluated in a descriptive manner by grouping a given characteristic in a range, as follows (Table 1):-From 0 to 5 for the broken edges of scales and the endocuticle (qualification for a given range was made on the basis of a comparison of the trait in question between different subjects and on the basis of the researcher’s own experience).-From 1 to 3 for the shape of the edge (1—convex, 2—straight, 3—concave); the evaluation was based on the researcher’s own experience and comparison between different photos.

Assessment of morphological lesions (pitting, oval indentations, rod-like macrofibrillar elements, globules, scratches, wavy edge) on the surface of scales was performed manually on images of the same size, usually 20 × 20 µm and 30 × 30 µm, to compare features between patients. At each location, 2–5 images were used for analysis.

### 2.6. Statistical Analysis

Statistical analysis was performed using *Statistica* software, v. 13.0 (TIBCO Software Inc., Kraków, Poland). Mean, minimal and maximal values along with the standard deviations were calculated for longitudinal parameters. 

## 3. Results

In this study, the hair surface was evaluated for three separate groups of parameters. The first subsection concerns metric measurements of hair scales. The measurements obtained are shown in Table 2. The second subsection focuses on mapping the hair in terms of morphological features of the natural process of hair delamination. The change in the distribution of these features with distance from the root of the hair is presented semi-quantitatively. The third subsection deals with selected non-characteristic morphological features on the surface of healthy hair. These changes were imaged in significantly greater numbers on the surface of hair from patients with diseases, i.e., lichen planopilaris and frontal fibrosing alopecia. However, since it also occurred on the hair of healthy subjects, it is presented descriptively above.

### 3.1. Dimensions and Size of the Scale

The hair scales were more-or-less rectangular in shape. The average length and width of the scales, depending on the distance from the root, differed only subtly. The measurements obtained are shown in Table 2. The main difference was observed in the shape of the free edge of each scale. Cells of intact hair show a smooth and convex contour. The edges of the cells, which wear down, become increasingly frayed and concave (Figure 2(a1–a4),b). The cells degenerate by breaking off small fragments from the free edge of the scale. This process is most intense in the center of the hair scales. The study using AFM captured moments of natural exfoliation of hair cells (Figure 2c).

### 3.2. Surface Phenomena Resulting from the Natural Delamination Process

In this study, four different scale’s surfaces differing in morphology were identified—striated surface, endocuticle, smooth surface and cortex. They were singled out as being key for a further evaluation of the advancement of the delamination process. They formed the basis of the morphological structure of the surface of each healthy hair. As the distance from the root of the hair increased, the individual surfaces were replaced by more hair, forming a kind of continuum of the process of natural wear and tear of hair cells (for maintaining the self-adaptive—disentangling—properties of the hair [2,3]) (Figure 2, Figure 3, Figure 4 and Figure 5).

The surface of the intact hair fiber, just near the end of the root, shows a longitudinal *striation*, which is a characteristic of the cuticle’s superficial layers (Figure 3(a1–a3)). With a greater distance from the root, such a surface is imaged less and less frequently, indicating gradual destruction of the hair. To visualize this surface, the probe should scan perpendicularly to the long axis of the hair. The striations were cylindrical in shape and parallel in arrangement, along the long axis of the hair. With a greater distance from the root end, they became flatter and began to transition into each other. On average, these striations were 300–450 nm wide and about 10 nm high. A detailed analysis of these striations, their individual differences and their association with scalp diseases will be included in a separate publication.

With a greater distance from the root end, fragments of the harder, outer layers of the cuticle (exocuticle and A-layer) exfoliate gradually and expose the surface of the *endocuticle* (Figure 3(b1,b2)). (Controlled detachment at the level of the soft endocuticle under friction occurs due to lipids—18-methyleicosanoic acid—that cover the outer surface of the cuticle cell [3]). The endocuticle is a granular, irregular surface. It was visible at the base of the free end of the scale, representing its remnant. It laid directly on top of the smooth surface of the scale.

The *smooth surface* is revealed when the outermost layers of the scales exfoliate. It represents cuticle structures such as the inner layer, the lower β-layer and the δ-layer. In this study, it was the surface on which various morphological lesions—scratches, pitting, globules—were more frequently observed. The smooth surface appeared proximal to the line delineating the ghost signs (see below). Viewing a particular scale (Figure 3(c1,c2)), one can see the following at a similar level: a striated surface of the distal scale that extends to the free end of the cell; a transverse line representing the transition of one scale into another (ghost signs); and a smooth surface as a remnant of the proximal scales (in the cephalad direction).

At the end tip of the hair, it was possible to visualize an irregular rod-shaped surface, which most likely corresponds to the hair’s *cortical cells* (Figure 3(d1,d2)). In the material studied, this type of cortex surface was found relatively rarely due to the fact that the scope of the study included the first 7 cm of each hair.

The scale exfoliates from the free end, initially showing its deeper layers until it disappears completely. Once the lower cell is exposed, a linear depression of the surface is visible, corresponding to the edge of the higher lying scale that has recently been destroyed. The cavity that is created is referred to as a *ghost sign* (Figure 2(d1–d4)). These ghost signs can be recognized because proximally from the transverse linear depression, the striations disappear and the smoother surface of the inner or lower β-layer is exposed. Ghosts are found in sections of hair, where the process of cuticle delamination begins. As the destruction of the cuticle progresses, ghosts also disappear.

With the distance from the root of the hair, the *free edges* of the cells became damaged. At first, the edges were smooth and undamaged, but gradually became increasingly frayed and broken. The *shape of the cells* also changed. At first, the free edges of the cells were rounded, convex. Then, they became concave, C-shaped.

Figure 6 graphically shows how the above features changed on the hair fibers of the control group subjects (S1–S10). It is possible to see how a particular feature disappeared or increased with distance from the root in a particular subject. You can also see individual differences in the intensity of delamination between subjects. It can be seen that some hairs (S4, S5) only started to show more delamination features towards the end of the hair section studied (so hair was less damaged and better cared for). However, one can see a repeatability and a trend in the formation of the hair fiber.

### 3.3. Other Morphological Phenomena Visualized on the Hair Surface

Pitting (Figure 4a,b)

One of the characteristic changes observed on all tested hairs was the finding of small, round excavations—pitting. In appearance, they resemble corrosion pits or pitting caused by friction in systems with grease (in the process of wear). In the healthy hair studied, they were regularly dispersed, did not form clusters on specific scales and were of similar size.

Oval indentations (Figure 4c,d)

The indentations differed from pitting in size and shape. They were about 10–12 times larger and oval in shape.

Scratches (Figure 4e,f)

Scratches and indentations of various shapes were observed along the entire length of the hair. In all samples, short but deeper scratches in the shapes of triangles, lines and commas were visible. In addition, we noted individual scratches along the long axis of the hair extending through several scales in the form of a precise line and with various other unexpected shapes. The lesions described above were less often seen on the striated surface and appeared more often on the deeper and softer layers of the cuticle.

Wavy edge (Figure 5a,b)

In individual samples, perfectly wavy edges of the scales were visible in some places. In addition, at a high magnification, the indentations on the surface of the scales in the ghost sign also take this shape. The cuticle cells with perfectly wavy edges in the proximal regions fracture along the long axis of the hair. As a result, linear cracks running from the base of the wave deep into the scales are visible.

Globules (Figure 5c,d)

The lesions were observed in various sections of the studied hairs in low quantities. The etiology of their formation is not entirely evident. They are probably related to mechanical damage to the hair surface. Globules appeared in the course of physiological delamination and exposing the lower cuticle layers. They were also observed in larger amounts in the immediate areas of sudden destruction of the hair fiber in patients with LPP (personal observation). Three different types of globules were visualized: (1) characteristic of delamination, flat elevation globules with dimensions on the order of hundreds of nanometers; (2) smaller globules (sand) with dimensions in the order of tens of nanometers (encountered less frequently), which were probably part of the cuticle layers underlying the endocuticle; (3) large conglomerates of contaminants lying on the surface of the hair with dimensions comprising thousands of nanometers, which were rather individual, found on the hair of specific patients, at different sections of the hair.

Rod-like macrofibrillar elements (Figure 5e,f)

There were clearly visible structures that were limited to a particular scale. Rod-like macrofibrillar elements were usually below the level of the surrounding cuticle cells. However, individual lesions were visualized at the level of other scales. They differed from the rods in the cortex layer in size and had a more regular, parallel formation.

**Figure 4 medicina-60-00297-f004:**
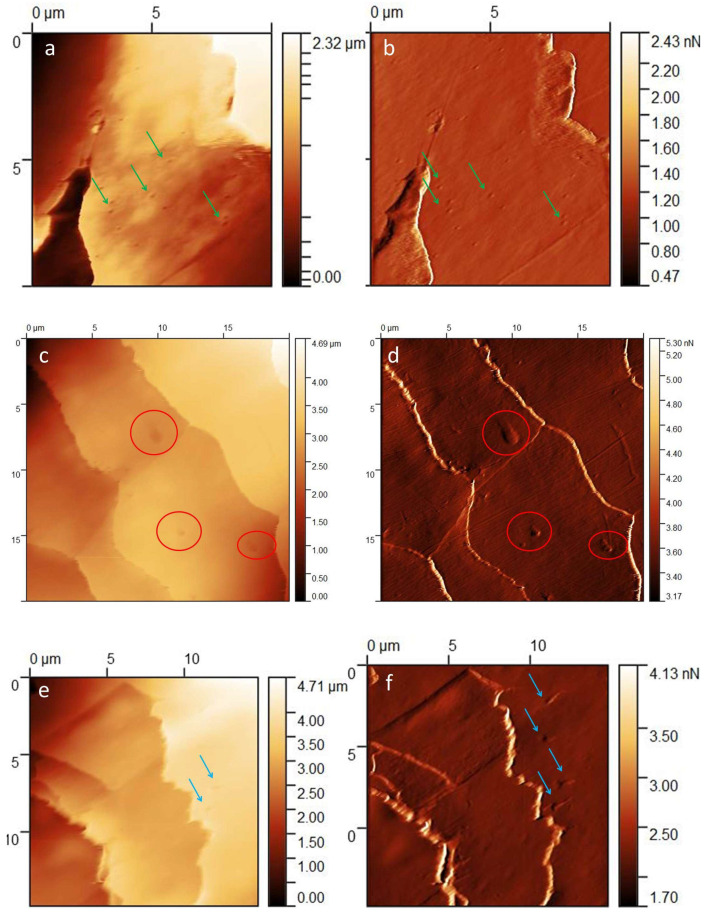
(**a**) Pitting—green arrows; *Z*-axis image. (**b**) Pitting—green arrows; deflection image. (**c**) Oval indentations—red circles; *Z*-axis image. (**d**) Oval indentations—red circles; deflection image. (**e**) Scratches—blue arrows; *Z*-axis image. (**f**) Scratches—blue arrows; deflection image.

**Figure 5 medicina-60-00297-f005:**
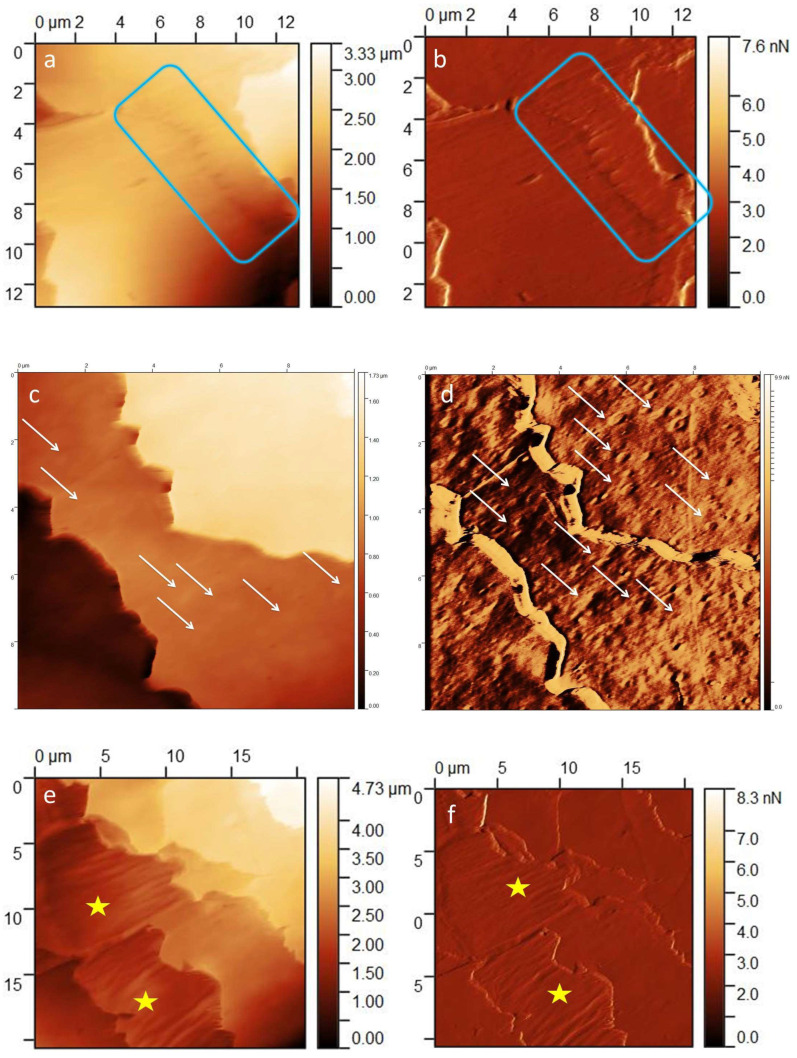
(**a**) Wavy edges—blue rectangle; *Z*-axis image. (**b**) Wavy edges—blue rectangle; deflection image. (**c**) Globules—white arrows; *Z*-axis image. (**d**) Globules—white arrows; deflection image. (**e**) Rod-like macrofibrillar elements—yellow star; *Z*-axis image. (**f**) Rod-like macrofibrillar elements—yellow star; deflection image.

**Figure 6 medicina-60-00297-f006:**
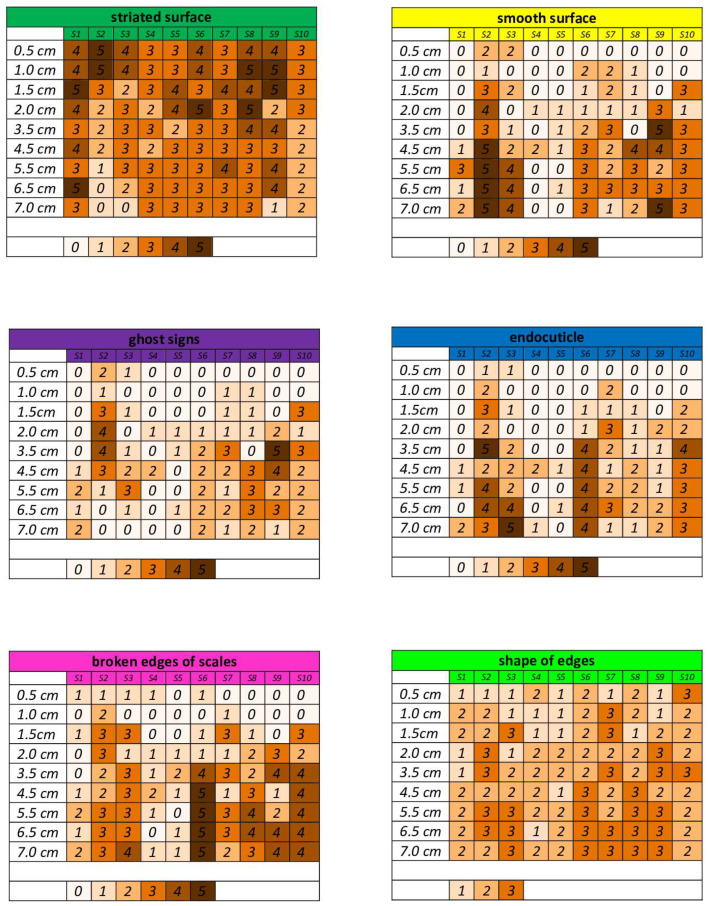
Distribution of delamination features on hair (parameters described in Table 1).

## 4. Discussion

### 4.1. Basics of AFM Operation and Functioning

AFM is a type of microscopy with a scanning probe (scanning probe microscopy—SPM). It is used to image atomic structure by scanning a sample with a sharpened tip mounted on a flexible cantilever. The technique allows imaging of samples in air, liquid and high vacuum. With the ability to capture images in a buffer solution, biological samples can be studied in their native states. It is possible to examine cells intravitally, without the need to stain, fix, dehydrate or label samples. This makes it possible to examine dynamic biological phenomena in a real-time manner, at the nanoscale [10,14,15]. In addition, by scanning the deflection of the cantilever, information from the Z-plane can be obtained, providing 3D images of the surface under examination. This enables us to perform a characterization of the morphology, thickness, surface roughness of the test sample and atomic forces, especially longitudinal forces [16]. In-plane AFM shows a resolution of about 1 nm, while for the surface of living cells it can approach ∼10 nm. Out-of-plane resolution is in the order of sub-nanometers. With AFM, measurements can be made at the micro- and nanoscale, and even atomistic events can be recorded at the edges of atomic lattice steps [15,17,18]. The scanning surface enables testing to be performed up to dimensions in the order of 100 µm in the horizontal plane and up to 10 µm in height [14].

The AFM is constructed from a laser, a cantilever with probe tip, a photodiode detector, a nanopositioning system with a piezoelectric scanner and a feedback imaging motion controller. The basis of operation with regard to this method concerns performing a scan with the tip of the probe and the surface of the test sample in the X and Y planes. By detecting the laser beam reflected from the top surface of the deflected cantilever, it is possible to measure the height of the sample in the Z-axis. The nanopositioning system is used to control the relative distance between the probe tip and the sample surface. Depending on the mode of operation, this system adjusts the deflection or oscillation of the cantilever to maintain the same spacing between the tip and the sample during the entire scanning process. Additional control feedback systems between the laser detector and scanner are essential for imaging performance and to avoid the destruction of biological samples [14,15,16,19]. Depending on whether the tip is in contact or almost in contact with the surface to be tested, AFM can operate in a contact mode, noncontact mode and in an intermediate mode (tapping mode). The AFM imaging mode depends on the characteristics of the samples being studied [17,18,19,20].

In *contact mode,* the AFM tip is constantly in contact with the surface to be analyzed. When imaging in contact mode with vertical forces of 10–30 nN, there is usually no cell damage. However, this method does not completely protect delicate biological samples from the deformation or degradation of their surfaces. It performs better with samples that are harder, rougher and that have a stiffer surface [14,21]. The vertical forces compress cell membranes and allow imaging of elements inside the cell. By examining cells without a cell wall, it is possible to image the cell nucleus or cytoskeleton network by increasing the vertical force. However, as the imaging force applied to the cells increases, the resolution of the images obtained decreases [14,22,23]. The force also provides information on the local viscoelasticity of the cell, adhesion forces and nanomechanical properties [24,25]. This is the most suitable mode for testing frictional forces [26]. Scanning can be performed in two modes of operation: constant force mode and constant height mode. The constant force mode provides information about the topography of the surface, considering the changes in the surface height of the sample. The constant height mode is primarily used for quick overview scanning of samples with small height differences [27].

In *noncontact mode*, where the scanner tip does not touch the surface to be tested, a high vacuum environment is required to examine biological samples. Consequently, this mode is not frequently applicable in medical research [20,27,28]. The noncontact mode enables us to detect the interaction forces between biomolecules. Attaching a specific molecule probe to the AFM tip makes it possible to find selected molecules of interest in the sample under study that specifically interact with the selected probe. Such a study is made possible via molecular recognition force microscopy (MRFM) or topography and recognition (TREC) microscopy [29,30,31,32]. In the noncontact mode, there is no possibility of testing the mechanical properties of the samples [21].

Given the limitations of the above methods, a *tapping mode* (also known as intermittent mode, dynamic mode) was constructed in 1993. This is the most suitable and most widely used method for testing delicate biological samples and polymers. In tapping mode, the tip cantilever typically oscillates between 100 and 300 kHz, with an amplitude of 30–100 nm. The tip moves toward the sample, touches it, and then moves away from the sample surface during each oscillation period [20,21,27,33,34,35]. The test can be conducted in any environment and achieves high resolution. Scanning in this mode is fast as in contact mode and, furthermore, there are no shear forces that are responsible for possible surface damage in contact mode. In this mode, topographic images and phase images are obtained [20,36]. The study of the viscoelastic properties, the variations in composition and adhesion is based on the measurement of the amplitude or phase shift between the free oscillation in air and the oscillation when the tip is tapped against the surface [20,21,37].

Recently, new mechanical imaging techniques based on the tapping mode have emerged, such as contact resonance, multifrequency AFM, multiharmonic mode AFM and viscoelastic mapping. These modes are used to study complex biomolecular and tissue systems due to their ability to provide multi-parametric, quantitative evaluations [26,38,39,40,41,42]. The most suitable for testing the mechanical properties of multicomponent coatings is *force mode*. In this mode, the surface to be tested moves in the direction of the tip. It shows a high resolution and does not damage the surface of samples [27].

AFM enables the creation of images of a sample’s surface under examination, providing information on 3D topography. In addition, monitoring cantilever deflection during scanning enables the measurement of the force acting between the tip and the sample [20]. Due to the different forces acting between the AFM tip and the sample surface (overlapping of electron orbitals or van der Waals forces), resultant repulsive or attractive forces are generated [20,27]. In contact mode, where the excursion force is repulsive, the tip is constantly in contact with the sample. In noncontact mode, the tip floats at a certain distance from the surface, and the force is attractive [14,16,20,21,27,40,43]. In tapping mode, the resultant force is alternately attractive or repulsive, while in the force mode, the force is initially attractive until the tip is so close to the sample that the resultant force becomes repulsive [20,27,40]. Probe sample mechanical interactions enable us to study nanomechanical, electrical, magnetic and tribological properties as well as surface mapping [16].

The AFM is a modern, advanced, multifunctional tool that allows for a whole range of measurements and tests. However, it also has several disadvantages. The main disadvantage is the speed of AFM imaging, which is relatively slow, making the AFM technique time-consuming. In addition, the measurement technique is generally complicated. To avoid damaging biological and delicate samples, the right mode, probe properties and controller parameters have to be chosen [15].

### 4.2. AFM Research on Hair

Previous research has focused on depicting healthy hair in material sciences. You and Yu (1997) were among the first to image human hair using AFM techniques. This was one of the largest studies available on quantitative changes in the hair surface under the influence of external environmental factors. For this purpose, accurate measurements of hair scale deviation were completed on a large control group. The surface of the intact hair imaged by You and Yu had a stepped pattern, where each scale overlapped each other. In order to evaluate the height (deviation) of the scales, they created line profiles, which are line graphs of the longitudinal section of the hair. The *x*-axis of this graph is the long axis of the hair, while the height difference on the surface of the hair is deposited on the *y*-axis, measured in the Z-plane of the scanner. The height of each step on the line graph represents the value of the scale’s deviation. This study showed that the height of each step in a profile line (scale deviation) varies greatly and changes even within a single hair fiber measured at different locations. Statistically, the step height of untreated hair statistically was 340.72 nm ± 90.57 nm (*n* = 210) [4]. In this study, the scale’s step height was relatively constant at approx. 480.5—531.1 nm. You and Yu also showed the dependence of scale deviation on pH and high temperature. In addition, this study described the changes occurring on the surface of heat-treated hair. The structural changes were mainly rough and uneven cuticle edges and granules [4]. In the following years, when step heights became a quantitative parameter for assessing hair, Smith (1998) developed a computational method for conducting rapid measurements from AFM-derived images. Based on 10 AFM images of brown European hair, thousands of line profiles were made in image analysis software (Topo-Metrix SPM Lab. 1996, Version 3.06.06, TopoMetrix Corporation, Santa Clara, CA, USA.). It was then possible to measure the cuticle step height using a computer method. The results obtained using the automated method coincided with those obtained using manual measurement. A wide distribution of results and similar step height values were obtained, similar to the study by You and Yu [5].

Swift et al. (2000) were the first to describe the overall morphology of the hair surface using AFM techniques. This is one of the most comprehensive and still current studies of the surface appearance of healthy hair. They found that the architecture of the hair surface changed gradually from the root to the tip. At 1 cm from the root, the scales were slightly curved and had a smooth edge. The surface of the scales themselves showed a slight undulation (striations), which developed when the hair cells passed through the inner root sheath. With distance from the root, small fragments broke off from individual scales causing the edges to be angularly serrated. As short as 3 cm from the root, most scales lost their smooth edge and granular remnants of exfoliated scales and exposed one of the inner layers of cuticular hair cells (endocuticle) appeared. At this distance from the root, the ghost sign was most often observed. Swift et al. explained that this is the imprint of a pre-existing scale, which has completely exfoliated, on the existing basal scale surface beneath it. With distance from the root, there was an increasing erosion of the scales until they completely disappeared, and the cortical surface was revealed near the tip of the hair. It had the appearance of longitudinally arranged rod-shaped elements. One of the main discoveries of their work was the imaging of striations on the surface of intact cuticle cells, which were not well visible using previous hair imaging methods (TEM, SEM). These occurred mainly at root ends. They had a parallel arrangement and comb shape. Each ridge was about 350 nm wide and up to about 9 nm high above the surface of the substrate [3]. The striated scaly surface has also been described in animal (mammalian) hair in other biological studies [44]. The above authors attributed this species-independent feature to all mammals. Swift et al. also studied the process of hair deterioration. They found that with external factors and dry friction, the hair cuticle delaminates at the cell boundary at the site of the so-called cell membrane complex (CMC). The new surface dominating the greater length of the hair has different chemical and physical properties. Additional signs of hair destruction included areas of exposed endocuticle in the form of granules covering 1/2 the thickness of a normal cuticle cell. These, however, are transient and quickly undergo complete exfoliation. In this work, Swift et al. distinguished between the two hair surfaces described above (with striations and the one formed after cuticular delamination). The difference in the angle of the two surfaces was also noted. These features may prove helpful in assessing the degree of hair damage [3].

Chen and Bhushan (2005) accurately imaged ghost signs on the hair surface and explained its genesis. In addition, they visualized the cuticle and cortex layers in cross sections [7].

The above morphological features of the surface of human hair were described in publications that aimed to demonstrate the feasibility of studying human hair with AFM. The authors did not assign specific properties to these features. They did not focus on the visual description of the hair and the importance of discrete morphological features such as pitting, globules or scratches. Although they are visible in some of their published photographs.

In contrast, Shin et al. (2012) studied the hair of patients, using AFM, with one of the most important dermatological diseases, psoriasis. They described morphological changes such as micropits and macropits. Micropits were described as indentations in diameter <0.5 µm or area <0.25 µm^2^, while macropits were described as indentations in diameter >0.5 µm or area >0.25 µm^2^. On hair taken from active psoriasis patches, the researchers found twice as many pits as on hair samples belonging to patients with psoriasis in comparison to unchanged skin. On the hair samples from active patches, the majority of pits were macropits, while micropits predominated over macropits on the hair from apparently unaltered skin. Micropits were also rarely observed in the control group. In addition, the hair of patients with psoriasis showed greater scale thickness and greater scale surface roughness. This was one of the few AFM studies of the hair surface in the field of skin and hair diseases. In addition, this study made it possible to visualize abnormalities at the nanoscale, as the same changes on the hair surface were found in patients with psoriasis—on the hairs taken from disease lesions and from apparently (clinically) healthy skin [45].

The studies cited above involved the imaging of the cuticle surface of human hair. This is still the most up-to-date research on this subject, as further work has begun to evolve into the study of nanomechanical properties, among other things. In this study, however, morphological features provide a reference point of what healthy hair is expected to look like. The characterization of hair on the basis of relatively well-imaged surface features can be helpful for comparing healthy hair with diseased hair.

In subsequent years, other studies have mainly focused on the effects of conditioners and hair care products on the hair damaged from cosmetic treatments. In addition, many studies have considered the assessment of tribological features, mechanical properties of hair cells and physical interactions on their surface [2,5,6,7,8,9,46]. Seshadri and Bhushan (2008) and Lu et al. (2019) studied virgin hair of different races and the hair damaged by cosmetic treatments, i.e., coloring or perming. Hair imaging was designed to assess the effectiveness of commercial conditioners and cosmetics [47,48,49].

The latest AFM studies of hair are being carried out mainly by physicists and engineers, with a trend away from simply imaging the surface of hair towards a deeper understanding of its structure, chemical composition and physico-mechanical properties. From 2020 to 2022, Fellows et al. used the AFM-IR technique to assess the chemical composition and chemical bonds of human hair. They showed the existing significant differences between the medulla, cortex and cuticle layer of human hair [50,51,52]. A few years earlier, Marcott et al. performed a similar but smaller study of the lipid composition of human hair using the AFM-IR technique [53]. In their study, Fellows et al. and Marcott et al. confirmed the great potential of combining infrared spectroscopy with the AFM technique to study complex biological samples.

## 5. Conclusions

This publication attempts to characterize the surface of healthy hair. On the basis of the authors’ own observations and the available literature, seven features were identified that may indicate the natural process of hair delamination (striated surface, endocuticle, smooth surface, cortex, ghost signs, broken edges of scales, shape of edges). The observation along the length of the hair of the above features made it possible to characterize the appearance of the reference hair in a descriptive and semi-quantitative way. The natural process of delamination is evidenced by the gradual breakaway of the scales and the change in shape of the edges from convex to concave. With detachment, the top layer of striated cuticle disappears, and the endocuticle and ghosts are gradually exposed, on average, at a distance of 2–5.5 cm from the root. With the distance from the tip root, in successive stages of delamination, the ghosts also disappear, and the surface of the endocuticle decreases until it disappears. Deeper smooth layers of the cuticle are exposed until the cortex at the end of the hair is uncovered. Based on the above observations, a preliminary conclusion can be drawn that the hairs’ delamination process between us is similar. This process can vary in intensity and its starting point (distance from the root).

Accurate measurements were taken of the length, width and height of the scaly deviation. The scale step height was relatively constant. The apparent scale length was maximal at the root end of the hair. The scale width, in contrast, was greatest at the tip end of hair. Both values changed only subtly. No significant differences were shown between individual subjects. However, the measurements were made on too small a group of subjects to perform a standardization of normal scale dimensions. To do so, measurements would have to be made in a group at least ten times larger.

While the results of the measurements do not have a significant impact on the assessment of healthy hair, subtle morphological changes on the surface of the hair scales may provide a direction for further research. In the physical sciences, such changes can be ignored. However, in medicine, clinical features and qualitative variables are more often given importance. Specifically, descriptive morphological changes are used in dermatology in examinations such as dermoscopy and reflectance confocal microscopy, for example. In the above publication, morphological lesions found on the surface of healthy hair, i.e., pitting, oval indentations, rod-like macrofibrillar elements, globules, scratches and wavy edges, are described. These features were visible in all healthy hairs, but in insignificant quantities. They occurred much more frequently on the hair of patients with scalp diseases (personal observation)—which will be a topic for future publications. Therefore, an attempt has been made to characterize morphological lesions at the nanoscale, which may be helpful in the early diagnosis of hair diseases in the future. The presented results will serve as a reference for the authors to conduct further research on the diseases of the scalp and hair (i.e., lichen planopilaris, frontal fibrosing alopecia and alopecia areata).

The main limitations of this study were its focus on morphological assessment of the hair surface only and its small sample size. Mechanical properties of the hair surface were not measured. The image capture time was 6 min at 256 points/line, 12 min at 512 points/line and about 20 min at 1024 points/line. Due to the time-consuming nature of the method and the need to take multiple measurements on the length of a single hair, a small group of subjects was studied. Patients selected for this study had not undergone hair treatments (i.e., hair coloring, keratin straightening or other treatments affecting the hair surface) in the last 3 months prior to this study. However, compliance with this requirement was based only on patients’ declarations. No other verification was possible. The selected patients clinically, and on a dermoscopic examination, did not present symptoms of scalp diseases. However, a subclinical course of a disease or a predisposition to develop a disease in the future, which could have affected the appearance of the hair at the nanoscale, could not be excluded. Moreover, the first 7 cm of each hair was examined. Due to the imaging of the cortex in some samples at this distance, further sections of hair were not examined. At greater distances from the end of the root, the hair structure is increasingly altered by external influences. In addition, the hair of patients with scalp diseases is short, and the results presented here were compared to a control group in a study of patients with lichen planopilaris and frontal fibrosing alopecia.

The limitations of the technique for working with AMF comprised the difficulty in scanning damaged areas of hair. In these areas, the AFM probe tip very often broke down due to unstable hair fragments that stuck to the probe. In addition, the diameter of the hair fiber decreased with distance from the root end. This caused a difficulty in positioning the probe tip over the hair, hitting it and centering the probe on the top of the hair. Consequently, each new point on which the probe landed was tested by performing pre-scans, which prolonged the procedure.

Future improvements to this study could include performing a nanomechanical characterization and investigating tribological properties of hair fibers. Parameters such as the roughness and stiffness of the hair scales and the elasticity of the hair fiber could be studied. This study could be improved by normalizing semi-quantitative variables in a more optimal manner and by taking more pictures and measurements. This could be possible by examining more patients, examining the hair along its entire length and taking measurements at smaller intervals. If the right equipment is available, it would be very interesting to combine nanoscale imaging techniques with the study of nanomechanical properties and spectrometric measurements.

## Figures and Tables

**Figure 2 medicina-60-00297-f002:**
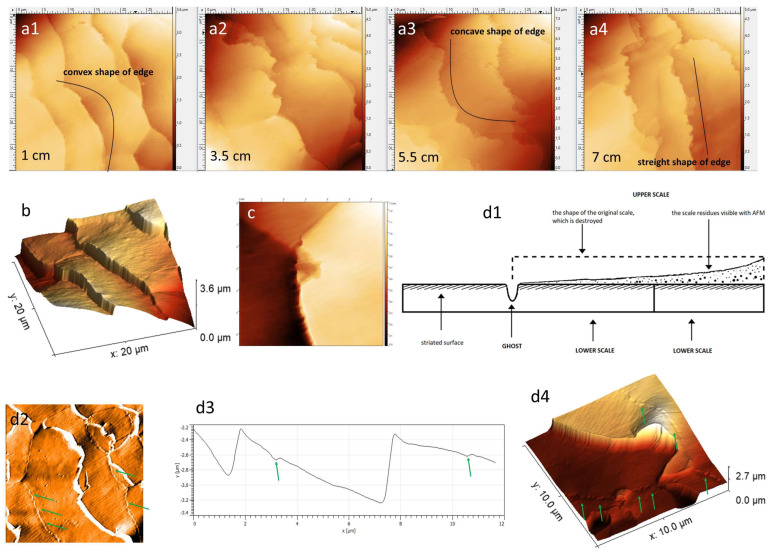
(**a1**–**a4**) The process of exfoliation of the scales depending on the distance from the root end of the hair—the black line marks the shapes of the scales’ edges; visible smooth (**a1**) and frayed (**a4**) edges of scales; *Z*-axis image. (**b**) Three-dimensional projection. (**c**) The beginning of the scale detachment; *Z*-axis image. (**d1**) Diagram of ghost sign formation. (**d2**) Ghost signs—green arrows; deflection image. (**d3**) Line profile of the scales with the cavity corresponding to the ghost sign marked with a green arrow. (**d4**) Ghost signs—green arrows; 3D projection.

**Figure 3 medicina-60-00297-f003:**
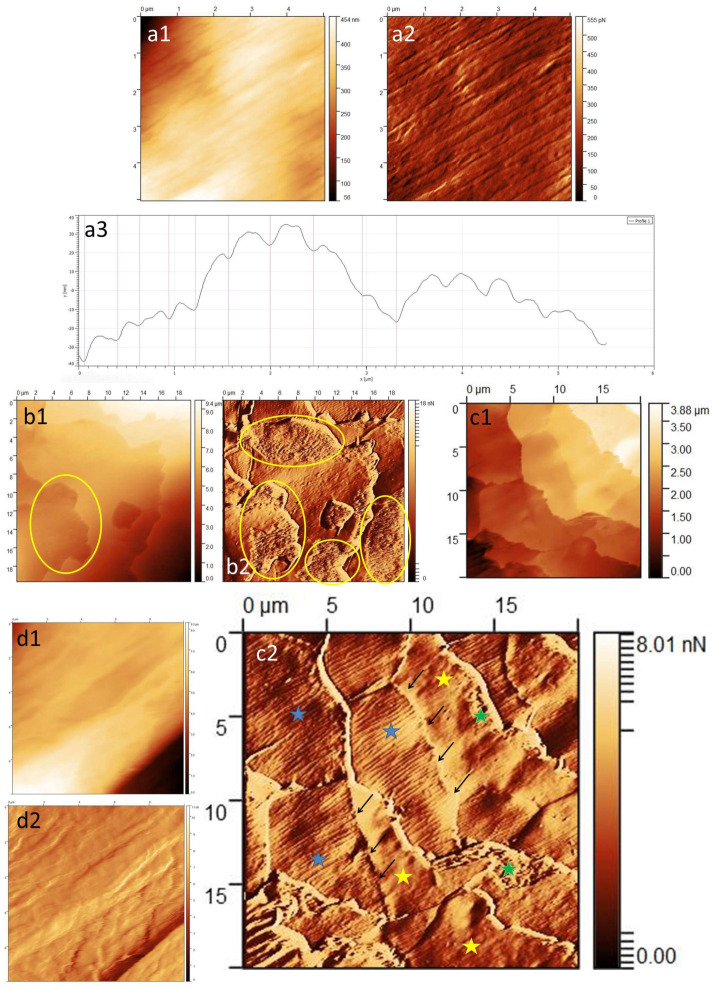
(**a1**) Striated surface; *Z*-axis image. (**a2**) Striated surface; deflection image. (**a3**) Striated surface; line profile. (**b1**) Endocuticle at the free edge of the scales—yellow circles; *Z*-axis image. (**b2**) Endocuticle at the free edge of the scales—yellow circles; deflection image. (**c1**) The appearance of the scales during the natural delamination process; *Z*-axis image. (**c2**) The appearance of the scales during the natural delamination process: striated scale (blue star), smooth scale (yellow star), ghost signs between the two scales (black arrows), endocuticle (green star); deflection image. (**d1**) Image of a fully exposed cortex—irregular, rod-shaped cortex surface; *Z*-axis image. (**d2**) Image of a fully exposed cortex—irregular, rod-shaped cortex surface; deflection image.

**Table 1 medicina-60-00297-t001:** Quantitative and descriptive representation of features indicating the natural process of hair delamination.

Feature	Absolute Number—Range or Qualitative Description of the Feature	Corresponding Grade
The number of scales with striated surface	0	0
	1–3	1
	4–7	2
	8–11	3
	12–15	4
	16–20	5
The number of scales with smooth surface	0	0
	1–2	1
	3–4	2
	5–6	3
	7–9	4
	10–12	5
The number of ghost signs	0	0
	1–2	1
	3–4	2
	5–6	3
	7–8	4
	9–10	5
Shape of edges	Convex	1
	Streight	2
	Concave	3
Broken edges	- Edge undamaged, smooth;	0
	- Edge smooth with single small cracks;	1
	- Numerous small cracks, smooth edge visible for less than 40%;	2
	- Lack of smooth edge, numerous larger cracks and scale breakages;	3
	- Deep breakouts, edge like chain saw’s teeth;	4
	- Very uneven, deep, different shaped breakouts.	5
Endocuticle	- absent;	0
	- Single nodules 2–4 nm in diameter, located at the base of the scales’ edges;	1
	- Single nodules spread over the entire scale surface;	2
	- Granular plaques 3–5 nm wide localized at the base of the scales’ edges;	3
	- Granular plaques 4–7 nm wide located across the width of the scales, at the base of the scale edges;	4
	- Granular structure occupying more than 50% of the cuticle surface.	5

**Table 2 medicina-60-00297-t002:** Measurements of the hair scales.

Distance from the Root	Scale Step Height—Mean [nm]	Scale Step Height—SD [nm]	Apparent Scale Length—Mean [nm]	Apparent Scale Length—SD [nm]	Scale Width—Mean [nm]	Scale Width—SD [nm]
0.5 cm	512.9	99.5	7879.5	1135.4	17,234.0	3649.5
1.0 cm	509.5	119.2	6847.4	1352.8	16,589.0	2245.0
1.5 cm	480.5	106.2	7562.5	1369.4	17,177.0	2616.2
2.0 cm	494.4	81.4	7673.4	1071.3	20,646.0	3685.4
3.5 cm	522.2	50.5	6719.1	1014.1	19,218.0	2865.8
4.5 cm	531.1	168.3	7855.9	1883.1	19,215.0	4010.3
5.5 cm	530.8	92.2	7495.0	1251.4	18,670.0	3236.4
6.5 cm	530.5	112.6	7255.7	1335.1	20,219.0	4404.2
7.0 cm	500.5	132.3	7589.2	1579.6	19,715.0	4310.5

## Data Availability

Data are available from the corresponding author upon reasonable request.

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
