# Peer review of "Evaluation of Surface Structure and Morphological Phenomena of Caucasian Virgin Hair with Atomic Force Microscopy"

_medicina, 2024, doi:10.3390/medicina60020297_

Round 1

Reviewer 1 Report

Comments and Suggestions for Authors

There are major issues that need to be amended, revised, and fixed. The comments and suggestions are as stated below. 

Abstract:

  1. The authors should highlight the main results of the study. 
  2. The general statement such as ‘AFM was developed in the late 1980s.’ should be omitted.
  3. Please revise the abstract according to the Medicina journal format. The current style does not follow the style of the journal.
  4. Please write the abstract with the standard style in order to represent the scope of study.

Introduction: 

  1. Please elaborate more on the background of the study and the improvement of this research compared to previous research. 
  2. The author should critically discuss the main focus of the study. The general information should be omitted in this section.
  3. The typo error such as ‘Baed’ should be revised. This applied to all the sections in the manuscript.

Results and Discussion:

  1. The discussion parts in this section are very shallow. The authors should include their critical opinion and related previous studies/references on state of the art in the manuscript.
  2. The characterization section is not well discussed. The authors only stated/mentioned the results obtained. The authors should discuss based on the previous studies/research (with examples) that have been conducted using the specific instrument for characterization. 
  3. The author should highlight/bold the subtopic title in the manuscript. The current flow of discussion is confusing and messy.
  4. The section for preparation and characterization of sample should be cited.
  5. The resolution of all the figures should be revised. The resolution is too low, and some fonts should be written with a suitable font style. 
  6. Please change the personal pronouns, such as ‘we’ in the text.

Conclusion: 

i.         Please separate the conclusion part from the discussion section. The conclusion section should be stand-alone in order to conclude the research study.

  ii.         The authors should conclude the main finding in this section.

 iii.         Describe the study's limitations and future improvement by including the data provided by the authors. 

References:

      i.         Some of the references need to be added in the text to support the explanation given.

     ii.         The references style does not follow the standard of the journal. Please revise it and cite recent articles to support the review articles as up-to-date research. 

Language Style: The English language quality is moderate and needs proofreading to meet the standard necessary for a scientific article. I would therefore recommend that the authors should look for the support of a native English speaker.

Comments on the Quality of English Language

Moderate editing of English language required.

Author Response

Responses to the reviewer: 

Reviewer: There are major issues that need to be amended, revised, and fixed. The comments and suggestions are as stated below. 

Abstract:

  1. The authors should highlight the main results of the study. 

Authors: Three main groups of parameters obtained were highlighted (metric measurements of hair scales, morphological features of the natural process of hair delamination and non-characteristic morphological features on the scales surface).

Reviewer: 

  1. The general statement such as ‘AFM was developed in the late 1980s.’ should be omitted.

Authors: The general statements have been removed.

 Reviewer: 

  1. Please revise the abstract according to the Medicina journal format. The current style does not follow the style of the journal.

Authors: The abstract has been corrected according to the format of the journal Medicina

Reviewer: 

  1. Please write the abstract with the standard style in order to represent the scope of study.

Authors: The style of the abstract was changed and the scope of the study was presented.

 Reviewer: 

Introduction: 

  1. Please elaborate more on the background of the study and the improvement of this research compared to previous research. 

Authors: The background of the study and factors that may indicate an improvement of this study compared to previous ones are reviewed.

Reviewer: 

  1. The author should critically discuss the main focus of the study. The general information should be omitted in this section.

Authors: The main aim of the study, which was to attempt to find such parameters of healthy hair that could be helpful in the non-invasive, molecular, rapid assessment of hair condition and its possible abnormalities, was discussed. The general information has been transferred to ‘Basics of AFM operation and functioning’ in the Discusion or has been deleted.

Reviewer: 

  1. The typo error such as ‘Baed’ should be revised. This applied to all the sections in the manuscript. 

Authors: Typo errors have been corrected.

 Reviewer: 

Results and Discussion:

  1. The discussion parts in this section are very shallow. The authors should include their critical opinion and related previous studies/references on state of the art in the manuscript.

Authors: The Discussion under the subtitle 'AFM research on hair' presents most of the available literature on the characterization of the surface of human hair by AFM. In addition, new trends in hair research such as the study of nanomechanical and chemical properties were presented.

Reviewer: 

  1. The characterization section is not well discussed. The authors only stated/mentioned the results obtained. The authors should discuss based on the previous studies/research (with examples) that have been conducted using the specific instrument for characterization. 

Authors: The Discussion presents the results obtained by previous researchers. The current results obtained were compared with those already published.

 Reviewer: 

  1. The author should highlight/bold the subtopic title in the manuscript. The current flow of discussion is confusing and messy.

Authors: All subtopics in the Materials and Methods, Results and Discussion have been bolded and highlighted.

 Reviewer: 

  1. The section for preparation and characterization of sample should be cited.???

Authors: A table was created (Tables 1) on quantitative and descriptive representation of features indicating the natural process of hair delamination. The results obtained are shown graphically in the accompanying Table 3.

In addition, s short paragraph on the sample preparation is included in the manuscript. As AFM does not require any specific hair sample preparation, this part is kept short as no specific preparation was done (native, untreated hairs were visualized). 

Reviewer: 

  1. The resolution of all the figures should be revised. The resolution is too low, and some fonts should be written with a suitable font style. 

Authors: The quality of the figures and style fonts have been improved.

Reviewer:

  1. Please change the personal pronouns, such as ‘we’ in the text.

Authors: The personal pronouns were removed.

 Reviewer: 

Conclusion: 

  1. Please separate the conclusion part from the discussion section. The conclusion section should be stand-alone in order to conclude the research study.

Authors: The conclusion section was separated from the discussion.

 Reviewer: 

  1. The authors should conclude the main finding in this section.

Authors: The main findings were concluded in this section.

Reviewer:

  1. Describe the study's limitations and future improvement by including the data provided by the authors. 

Authors: The study's limitations and future improvement were described in the conclusion

Reviewer: 

References:

  1. Some of the references need to be added in the text to support the explanation given.

Authors: Some references have been added to the text in the results and discussion.

Reviewer: 

  1. The references style does not follow the standard of the journal. Please revise it and cite recent articles to support the review articles as up-to-date research. 

Authors: References style was revised. The most up-to-date research on the subject was cited.

Reviewer: 

Language Style: The English language quality is moderate and needs proofreading to meet the standard necessary for a scientific article. I would therefore recommend that the authors should look for the support of a native English speaker.

Authors: WE have carefully reviewed the entire manuscript and corrected the typo, spelling and grammar mistakes. 

Reviewer 2 Report

Comments and Suggestions for Authors

The authors evaluated the surface structure and morphological phenomena of hair by AFM. This is a very basic project done by AFM, which is quite rough, but maybe publishable after polishing. Please refers to the following issues:

Major:

1. The authors use Gwyddion to treat the AFM image. It is not acceptable to just give a screenshot of the software windows. Please plot the figures carefully, and export the gywddion screenshots into professional AFM images.

Minor:

2. What is the frequency of the cantilever? 

3. What is the resolution of the image, i.e., what is the pixels?

4. Please note the scanning velocity.

Comments on the Quality of English Language

Minor editing of English language required

Author Response

Reviewer: 

Major:

  1. The authors use Gwyddion to treat the AFM image. It is not acceptable to just give a screenshot of the software windows. Please plot the figures carefully, and export the gywddion screenshots into professional AFM images.

Authors: Using Gywddion, files from Nanosurf CX ver 3.10.3.7 were exported to professional images saved as JPGs.

Reviewer: 

Minor:

  1. What is the frequency of the cantilever? 

Authors: Imaging was performed in contact mode using PPP-FMAuD-10 tips (NANOSENSORS™) mounted on cantilevers of force constant 1.9 N/m and 75 kHz frequency.

Reviewer: 

  1. What is the resolution of the image, i.e., what is the pixels?

Authors: The resolution of the images was 512-1024 points/line.

Reviewer: 

  1. Please note the scanning velocity.

 Authors: The scanning velocity was 0.5 s/line.

Reviewer: Comments on the Quality of English Language

Minor editing of English language required

Authors: We have carefully reviewed the entire manuscript and corrected the spelling and grammar mistakes. 

Round 2

Reviewer 1 Report

Comments and Suggestions for Authors

Please remove the 'Conclusions:' wording in the abstract section.

Comments on the Quality of English Language

Minor editing of the English language is required.

Author Response

We have removed the "Conclusions" wording from the abstract, although we were requested previously to add it by the editorial team. 

Reviewer 2 Report

Comments and Suggestions for Authors

I believe this work can be accepted.

Author Response

We are grateful to the reviewer for his/her supportive words.